# Learning Adaptive Topology with FiLM-Guided Distillation for Tertiary Structure-Based RNA Design

**Zixun Zhang** [1 2 3] **Yuncheng Jiang** [4] **Yuzhe Zhou** [1 2 3] **Jiayou Zheng** [1 2 3] **Shuguang Cui** [2 1 3] **Zhen Li**[✉ 2 1 3]

## Abstract

Tertiary structure-based RNA design aims to generate RNA sequences that can fold into desired 3D structures, but remains a challenging problem due to the scarcity of annotated data, structural noise, and the intrinsic complexity of RNA topology. Existing structure-to-sequence frameworks largely rely on static k-nearest neighbor graphs and rigid message passing schemes, which fail to capture the flexible and heterogeneous nature of RNA geometry. To address these issues, we propose a unified framework, ATL-FGD, that integrates Adaptive Topology Learning (ATL) and FiLM-Guided Distillation (FGD) for robust RNA design. ATL introduces a differentiable edge gating mechanism to jointly learn topology and representation, enabling the model to construct data-driven, layer-adaptive graphs that better reflect structural dynamics and biochemical consistency. On top of this, FGD bridges structural and sequence representations via feature-wise linear modulation, softly transferring the semantic knowledge from RNA foundation models without relying on them during inference. Extensive experiments on tertiary structure-based RNA design benchmarks demonstrate that our approach achieves significant improvements in both sequence recovery and structural fidelity.

## 1. Introduction

RNA molecules serve as versatile functional elements underlying diverse biological processes, including gene regu-

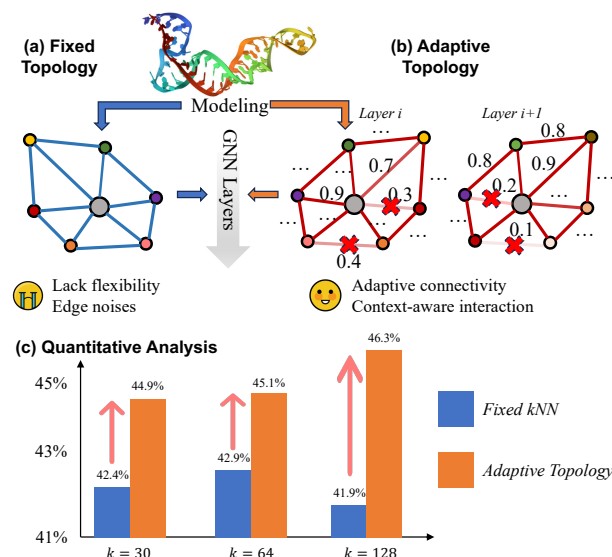

*Figure 1.* Overview of fixed compared with our adaptive topology modeling. (a) Conventional fixed $k$NN graphs impose static connectivity, which often introduces edge noise and lacks flexibility to reflect structural variation. (b) Our proposed adaptive topology learning dynamically adjusts the graph structure at each GNN layer through learnable edge gating, enabling context-aware and biochemically consistent message passing. (c) Quantitative analysis shows that adaptive topology consistently improves performance across different $k$ settings, alleviating the limitations.

lation, catalysis, molecular recognition, and signal transduction (Morris & Mattick, 2014; Breaker, 2012). In contrast to proteins, whose folded structures are primarily stabilized by hydrophobic packing, RNAs exhibit intrinsic conformational flexibility governed by hydrogen bonding, base-pairing geometry, and electrostatic interactions (Leontis & Westhof, 2001; Tinoco Jr & Bustamante, 1999). Consequently, designing RNA sequences that fold into a desired tertiary conformation, termed tertiary structure-based RNA design, constitutes a fundamental yet formidable inverse problem in computational biology. A robust solution would enable de novo creation of catalytic ribozymes, RNA switches, and therapeutic aptamers (Paige et al., 2011; Ellington & Szostak, 1990), unlocking transformative potential for synthetic biology and RNA-based therapeutics. Nevertheless, the combinatorial explosion of nucleotide in-

[1]Shenzhen Future Network of Intelligence Institute (FNii-Shenzhen), Shenzhen, China [2]School of Science and Engineering (SSE), the Chinese University of Hong Kong, Shenzhen (CUHK-Shenzhen), Shenzhen, China [3]Guangdong Provincial Key Laboratory of Future Networks of Intelligence, Shenzhen, China [4]West China Hospital, Sichuan University, Chengdu, China. Correspondence to: Zhen Li <lizhen@cuhk.edu.cn>.

*Proceedings of the $43^{rd}$ International Conference on Machine Learning*, Seoul, South Korea. PMLR 306, 2026. Copyright 2026 by the author(s).

teractions and the scarcity of experimentally resolved RNA 3D structures render this task substantially more challenging than its protein counterpart.

Recent advances in geometric deep learning have enabled protein-style design paradigms to be extended to tertiary structure-based RNA design (Dauparas et al., 2022; Gao et al., 2023). Most learning-based methods cast RNA inverse folding as conditional generation on a 3D structure graph: nucleotides are nodes, and edges encode geometric relations (e.g., distances and angles), followed by message passing to produce sequence distributions (Tan et al., 2024; 2025). To better capture 3D geometry, equivariant architectures further incorporate directional features via GVP-style representations (Jing et al., 2021; Wong et al., 2024; Joshi et al., 2025). Complementary to architectural advances, recent efforts explore learning objectives and generative formulations that improve data efficiency and robustness, including hierarchical contrastive learning (Tan et al., 2024), multi-conformation consistency learning for RNA flexibility (Joshi et al., 2025), and diffusion-based generation to enhance diversity and geometric coherence (Huang et al., 2024; Nori & Jin, 2024).

Despite notable progress, a key bottleneck remains the reliance on static graph construction, which imposes a hand-crafted and globally fixed topology on a molecule whose effective interactions are heterogeneous and context-dependent. Most approaches build $k$-nearest-neighbor ($k$NN) or radius graphs with a fixed $k$ (Tan et al., 2024; 2025; Joshi et al., 2025). In principle, enlarging $k$ expands the receptive field and should facilitate long-range information aggregation, however, RNA tertiary structures exhibit pronounced geometric heterogeneity, where non-canonical base pairs, pseudoknots, and long-range tertiary motifs jointly shape the fold (Leontis & Westhof, 2001; Tinoco Jr & Bustamante, 1999). Consequently, increasing $k$ introduces biochemically irrelevant edges that inject structured noise into message passing, leading to saturated or even degraded performance as empirically observed in Fig. 1. This issue is further amplified in the low-data regime of RNA structural supervision, where performance can become sensitive to graph construction choices and other hand-crafted priors, in contrast to the more data-rich protein setting.

To address these limitations, we propose Adaptive Topology Learning with FiLM-guided Distillation, ATL-FGD, a unified framework that jointly improves topology modeling and representation learning for tertiary structure-based RNA design. ATL-FGD treats graph topology as learnable rather than static, enabling dynamic adjustment of node connectivity at each GNN layer via differentiable edge gating. This mechanism mitigates noise from fixed $k$NN graphs while capturing long-range, context-dependent biochemical interactions fundamental to RNA tertiary geometry. Given that

RNA foundation models trained on large-scale sequence corpora encode rich contextual and functional priors, we introduce FiLM-guided Distillation to inject such knowledge into the structural learning process. Specifically, FiLM-guided Distillation leverages these priors by allowing both the teacher and student representations to generate scale and bias factors through FiLM layers, and then aligns these factors during training enables semantic transfer from teacher to student representations. This mechanism provides a soft modulation pathway rather than direct feature alignment that mitigates data scarcity while eliminating the need for the teacher model during inference. Together, these components enable flexible, biochemically consistent, and data-efficient representations, yielding substantial performance gains.

In summary, our contributions are three-fold: (1) We propose Adaptive Topology Learning with FiLM-guided Distillation, ATL-FGD, a unified framework for tertiary structure-based RNA design that jointly addresses static graph topology construction and data scarcity. (2) Within this framework, we introduce an adaptive topology learning mechanism that dynamically optimizes RNA graph connectivity to capture flexible and heterogeneous structural dependencies, alongside a FiLM-guided distillation strategy that transfers semantic priors from RNA foundation models to enhance structure-aware representation learning. (3) Extensive experiments on benchmark datasets demonstrate that ATL-FGD consistently improves both sequence recovery and structural fidelity, outperforming state-of-the-art baselines and validating the effectiveness of our design.

## 2. Related Works

### 2.1. RNA Design

RNA sequence design has long been a crucial problem in computational biology, with applications in synthetic biology, therapeutics, and nanotechnology. Classical inverse folding approaches, such as RNAfold (Lorenz et al., 2011), Mfold (Zuker, 2003), RNAInverse (Hofacker et al., 1994), and NUPACK (Zadeh et al., 2011), relied on thermodynamic energy minimization and combinatorial search to design sequences folding into target secondary structures. With the rise of deep learning, a new generation of data-driven and geometry-aware RNA design frameworks has emerged. Graph-based models such as RDesign (Tan et al., 2024) and R3Design (Tan et al., 2025) adopted MPNN backbones for tertiary structure encoding, while gRNAde (Joshi et al., 2025) and RiboDiffusion (Huang et al., 2024) incorporated geometric vector perceptrons and diffusion-based conditioning for flexible and geometry-coherent generation. RhoDesign (Wong et al., 2024) further validated structure-informed design experimentally for light-up aptamers. In addition, RNAFlow (Nori & Jin, 2024) and RiboFlow (Ma et al., 2025) introduced flow matching to jointly co-design

RNA sequences and structures. Overall, these studies collectively mark a paradigm shift from energy-based inverse folding to geometry- and representation-driven RNA design. Yet, despite architectural and algorithmic advances, most methods have paid limited attention to graph topology modeling, typically constructing graphs based on fixed $k$NN or radius. Such rigid topologies fail to capture RNA's flexibility and context-dependent connectivity, thereby motivating our proposed adaptive topology learning framework.

## 2.2. Graph Modeling of Biomolecular Structures

Graph neural networks (GNNs) have become a widely used framework for modeling biomolecular structures, whose interactions are naturally graph-structured. Proteins and RNAs are often represented as residue-/nucleotide-level graphs, where nodes denote residues or nucleotides and edges encode spatial proximity, physicochemical interactions, or experimentally derived contacts. By iterative message passing, GNNs aggregate information over local neighborhoods to capture longer-range dependencies relevant to structural stability and biological function. Early studies applied GNNs to inverse folding by conditioning sequence generation on backbone or contact graphs (Ingraham et al., 2019), establishing the viability of learning sequence-structure mappings from graph representations. To improve geometric fidelity, later work introduced explicit 3D inductive biases and symmetry-aware designs. The Geometric Vector Perceptron (GVP) (Jing et al., 2021) jointly updates scalar and vector features to enable orientation-aware message passing, while EquiPNAS (Roche et al., 2024) enforces E(3)-equivariance and integrates protein language model embeddings for structure and binding-site prediction. Beyond representation learning, graph-based methods have also advanced functional modeling: GraphSite (Shi et al., 2022) identifies ligand-binding pockets, GraphEC (Song et al., 2024) predicts enzyme functions from geometric graphs, and DeepRank-GNN (Réau et al., 2023) captures protein-protein interface patterns from structural data.

## 2.3. Cross-Modal Distillation

Cross-modal knowledge distillation (KD) is an effective paradigm for transferring semantic priors across heterogeneous modalities. In 3D perception, CMKDNet (Hong et al., 2022) pioneered LiDAR-to-camera distillation, and X3KD (Klingner et al., 2023) unified multi-modal, multi-task, and multi-stage transfer for improved camera-based 3D detection. LabelDistill (Kim et al., 2024) and VeXKD (Ji et al., 2024) further enhanced robustness via label-guided supervision and BEV-integrated fusion, respectively. Beyond vision, cross-modal KD has been extended to broader sensory and language domains. Visual-to-EEG Distillation (Zhang et al., 2022) transferred affective representations across perception channels, while the Modality Focusing

Hypothesis (Xue et al., 2022) analyzed representational overlap that enables effective transfer. XKD (Sarkar & Etemad, 2024) and C2KD (Huo et al., 2024) addressed domain alignment and adaptability through self-supervised reconstruction and sample-adaptive distillation. In vision language domain, Align-KD (Feng et al., 2025) distilled cross-modal alignment from large VLMs into compact students, and xMOD (Lahlali et al., 2025) enabled unsupervised 2D-to-3D transfer by leveraging motion cues. These studies highlight cross-modal KD as a general bridge between complementary modalities. In this work, we extend this principle to the biomolecular domain, transferring RNA foundation model priors into structure-based graph learning for efficient and data-adaptive RNA sequence design.

## 3. Methodology

Tertiary structure-based RNA design aims to generate an RNA sequence $\mathbf{Y} = (y_1, \ldots, y_L)$ with $y_i \in \{\mathrm{A, U, C, G}\}$ that is compatible with a given RNA tertiary structure $S$. We represent $S$ by atomic coordinates $\mathbf{X} \in \mathbb{R}^{L \times n \times 3}$, where $L$ is the sequence length and $n$ denotes the number of representative atoms per nucleotide. Following RDesign (Tan et al., 2024), we use six backbone atoms $\{\mathrm{P, O5', C5', C4', C3', O3'}\}$, i.e., $n = 6$.

We model the input structure as an attributed graph $G = (V, E)$, where each node corresponds to a nucleotide and is associated with atom-derived geometric features, and edges are constructed based on spatial proximity. A stack of GNN layers encodes $G$ into node embeddings $\{h_i\}_{i=1}^{L}$, which are mapped by a position-wise prediction head to nucleotide distributions, yielding the designed sequence. Despite its effectiveness, this standard pipeline is limited by (i) static graph construction, which may introduce noisy or irrelevant edges under fixed $k$NN priors, and (ii) the scarcity of labeled RNA 3D structures for supervised representation learning.

To address these issues, we propose Adaptive Topology Learning with FiLM-Guided Distillation (ATL-FGD), which consists of two core modules. Adaptive Topology Learning treats graph connectivity as learnable and performs layer-wise edge gating to adapt the effective neighborhood to each structure, thereby reducing noise from fixed $k$NN graphs while capturing context-dependent interactions in RNA tertiary geometry. FiLM-Guided Distillation injects semantic priors from RNA foundation models into the structure encoder via Feature-wise Linear Modulation (FiLM) (Perez et al., 2018) improving data efficiency. Fig. 2 illustrates the overall pipeline.

### 3.1. Adaptive Topology Learning

Rather than fixing graph connectivity, Adaptive Topology Learning (ATL) enables the encoder to learn which candi-

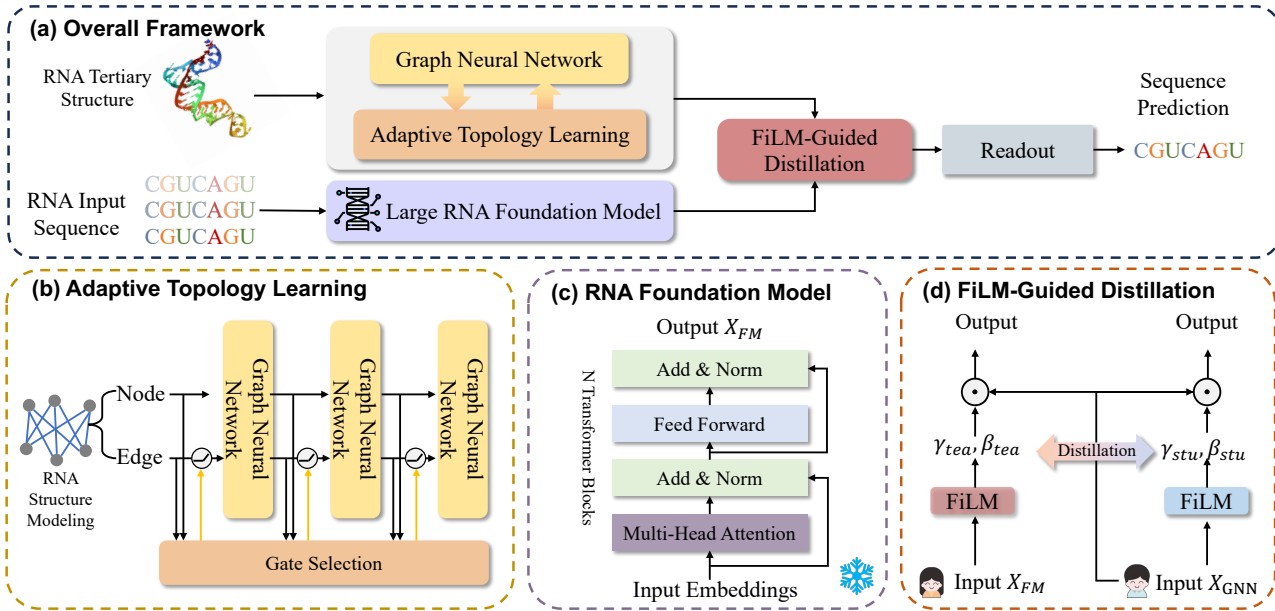

*Figure 2.* Overview of ATL-FGD, which integrates Adaptive Topology Learning (ATL) and FiLM-Guided Distillation (FGD) for accurate and generalizable RNA tertiary structure–based sequence design.

date edges should participate in message passing at each GNN layer. In doing so, the topology of the working graph becomes a learnable, layer-dependent subgraph of the input graph, allowing the model to suppress noisy edges while retaining structure-relevant contacts.

Formally, given an input graph $G = (V, E)$ at layer $\ell$, we infer a masked edge set $\tilde{E}^{(\ell)}$ via a binary mask $\tilde{m}_{ij}^{(\ell)} \in \{0, 1\}$ for each candidate edge $(i, j) \in E$. We first compute an edge representation by concatenating the current node states and raw edge features:

$$r_{ij}^{(\ell)} = \mathrm{concat}\big(h_i^{(\ell)}, h_j^{(\ell)}, e_{ij}\big),$$
$$p_{ij}^{(\ell)} = \sigma\big(f_\theta(r_{ij}^{(\ell)})\big),$$

where $f_\theta(\cdot)$ is a lightweight MLP and $\sigma(\cdot)$ is the sigmoid function. The score $p_{ij}^{(\ell)} \in (0, 1)$ parameterizes the edge activation probability conditioned on the current node context.

**Stochastic Gating for Training.** During training, we sample a Bernoulli gate and apply a straight-through estimator to enable end-to-end optimization:

$$u_{ij} \sim \mathcal{U}(0, 1),$$
$$m_{ij}^{(\ell)} = \mathbf{1}\{u_{ij} < p_{ij}^{(\ell)}\},$$
$$\tilde{m}_{ij}^{(\ell)} = m_{ij}^{(\ell)} + p_{ij}^{(\ell)} - \mathrm{stopgrad}\big(p_{ij}^{(\ell)}\big).$$

This yields discrete edge selection in the forward pass while using surrogate gradients through $p_{ij}^{(\ell)}$ in the backward pass, encouraging exploration over candidate topologies with stable optimization.

**Deterministic Gating for Inference.** At inference time, we replace the stochastic sampling with deterministic rule:

$$\tilde{m}_{ij}^{(\ell)} = \mathbf{1}\{p_{ij}^{(\ell)} > \tau\},$$

where $\tau$ is set to $0.5$ by default, producing a sparse and stable working subgraph without sampling noise.

After obtaining the $\tilde{m}_{ij}^{(\ell)}$, message passing at layer $\ell$ is performed on the masked graph $\tilde{E}^{(\ell)} = \{ (i, j) \in E \mid \tilde{m}_{ij}^{(\ell)} = 1 \}$ (equivalently, using masked edge features $\tilde{m}_{ij}^{(\ell)} \cdot e_{ij}$). Since the mask is recomputed at every layer, ATL learns layer-adaptive neighborhoods, enabling different layers to emphasize different structural cues. Alg. 1 summarizes the procedure.

### 3.2. FiLM-Guided Distillation

To alleviate data scarcity, we propose FiLM-Guided Distillation (FGD) to transfer contextual priors from an RNA foundation model into the structure encoder. A naive choice is to directly align teacher and student features, however, this is often ill-suited due to a receptive-field mismatch between the two representations. RNA-FM embeddings are produced by self-attention, where each token integrates sequence-wide context, whereas structure-based embeddings are obtained by local message passing over geometric neighborhoods. Consequently, pointwise feature matching can suffer from semantic mismatch and yield weak supervision. FGD addresses this by distilling knowledge through a modulation interface instead of direct feature alignment. Specifically, we map both teacher and student representa-

**Algorithm 1** Adaptive Topology Learning (ATL) at Layer $\ell$

---

**input** Graph $G = (V, E)$ with node states $\{h_i^{(\ell)}\}_{i=1}^L$ and edge features $\{e_{ij}\}_{(i,j)\in E}$; scorer $f_\theta$; GNN layer GNN$(\cdot)$ at layer $\ell$; threshold $\tau$ (default 0.5); mode $\in \{\text{TRAIN}, \text{EVAL}\}$.

**output** Updated node states $\{h_i^{(\ell+1)}\}_{i=1}^L$.

    **for** each edge $(i,j) \in E$ **do**
        $r_{ij}^{(\ell)} \leftarrow \text{concat}\big(h_i^{(\ell)}, h_j^{(\ell)}, e_{ij}\big)$
        $p_{ij}^{(\ell)} \leftarrow \sigma\big(f_\theta(r_{ij}^{(\ell)})\big)$
    **end for**
    **for** each edge $(i,j) \in |E|$ **do**
        **if** mode = TRAIN **then**
            $u_{ij} \sim \text{Uniform}(0,1)$
            $m_{ij}^{(\ell)} \leftarrow \mathbf{1}\{u_{ij} < p_{ij}^{(\ell)}\}$
            $\tilde{m}_{ij}^{(\ell)} \leftarrow m_{ij}^{(\ell)} + p_{ij}^{(\ell)} - \text{stopgrad}(p_{ij}^{(\ell)})$
        **else**
            $\tilde{m}_{ij}^{(\ell)} \leftarrow \mathbf{1}\{p_{ij}^{(\ell)} > \tau\}$
        **end if**
    **end for**
    $\{h_i^{(\ell+1)}\}_{i=1}^L \leftarrow \text{GNN}\Big(\{h_i^{(\ell)}\}_{i=1}^L, \{\tilde{m}_{ij}^{(\ell)} \cdot e_{ij}\}_{(i,j)\in E}\Big)$
    **return** $\{h_i^{(\ell+1)}\}_{i=1}^L$

---

tions to feature-wise FiLM parameters (scale and bias), and align these parameters during training. Intuitively, the student learns to reproduce the teacher-induced modulation behavior, benefiting from global semantic priors while keeping its representation grounded in 3D structure.

Formally, given a frozen teacher encoder $E_{\text{tea}}$ and a student GNN encoder $E_{\text{stu}}$, we obtain both representations with ground truth sequence $\mathbf{Y}^{\text{true}}$ and structure graph $G$,

$$h_{\text{tea}} = E_{\text{tea}}(Y), \ h_{\text{stu}} = E_{\text{stu}}(G).$$

We then apply FiLM heads $g_{\text{tea}}$ and $g_{\text{stu}}$ to produce modulation parameters:

$$(\gamma_{\text{tea}}, \beta_{\text{tea}}) = g_{\text{tea}}(h_{\text{tea}}), \ (\gamma_{\text{stu}}, \beta_{\text{stu}}) = g_{\text{stu}}(h_{\text{stu}}),$$

where $\gamma$ and $\beta$ are feature-wise scale and bias factors. The student representation is modulated as

$$\hat{h}_{\text{stu}} = \gamma_{\text{stu}} \odot h_{\text{stu}} + \beta_{\text{stu}},$$
$$\tilde{h}_{\text{stu}} = \gamma_{\text{tea}} \odot h_{\text{stu}} + \beta_{\text{tea}},$$

using the student- and teacher-induced FiLM parameters, respectively. The modulated representations are then fed to the prediction head to produce nucleotide distributions and optimized with the standard cross-entropy loss.

During training, we align the student FiLM parameters to the teacher targets with an MSE objective,

$$\mathcal{L}_{\text{FGD}} = \|\gamma_{\text{stu}} - \gamma_{\text{tea}}\|_2^2 + \|\beta_{\text{stu}} - \beta_{\text{tea}}\|_2^2,$$

which provides a stable distillation signal without enforcing direct feature matching. At inference time, we remove the teacher branch and retain only the student FiLM pathway, i.e., $(\gamma_{\text{stu}}, \beta_{\text{stu}})$ are generated solely from $h_{\text{stu}}$.

### 3.3. Training Objective

During training, we jointly optimize the model with a sequence recovery objective and the FiLM-guided distillation loss $\mathcal{L}_{FGD}$. The sequence recovery objective supervises nucleotide prediction via the standard cross-entropy loss. Given the ground-truth sequence $\mathbf{Y}^{\text{true}} = (y_1^{\text{true}}, \ldots, y_L^{\text{true}})$, and the predicted probability $P(y_i)$ at each position $i$

$$\mathcal{L}_{task} = -\sum_{i=1}^L \log P\big(y_i = y_i^{\text{true}}\big).$$

The distillation loss $\mathcal{L}_{FGD}$ is computed on the final-layer student representation $h_{\text{stu}}$, aligned with the corresponding teacher-induced FiLM factors. The teacher representation $h_{\text{tea}}$ is obtained from a frozen pretrained RiNALMo 650M RNA foundation model (Penić et al., 2025). The overall loss combines both objectives,

$$\mathcal{L}_{total} = \mathcal{L}_{task} + \alpha \cdot \mathcal{L}_{FGD},$$

where $\alpha$ balances the contribution of the distillation loss. The teacher branch is used only during training to generate FiLM factors, and is removed at inference with no access to sequence information.

## 4. Experiment

### 4.1. Dataset and Implementation Details

We implement ATL-FGD in PyTorch and follow the same geometric preprocessing as RDesign. All structures are collected from RNASolo, and we adopt two benchmark splits: (i) the RDesign split version and (ii) the gRNAde single-state split version. Following the RDesign evaluation protocol, we additionally report generalization performance on two generalization benchmarks, Rfam (Gardner et al., 2009; Nawrocki et al., 2015) and RNAPuzzle (Miao et al., 2020). We compare ATL-FGD with several representative baselines, including StructMLP, StructGNN, GraphTrans (Ingraham et al., 2019), RDesign (Tan et al., 2024), gRNAde (Joshi et al., 2025), and R3Design[1] (Tan et al., 2025).

ATL-FGD adopts the same MPNN backbone as RDesign with 6 layers, and integrates the Adaptive Topology Learning module into each MPNN layer. We train the model using AdamW with an initial learning rate of $1 \times 10^{-3}$, and cosine annealing for 200 epochs. All experiments are conducted on a single NVIDIA A100 (80GB), and results are

---

[1]We re-implement the results of R3Design and remove the iterative generation for fair comparison.

*Table 1.* The overall performance on RDesign split. The best results are highlighted in bold.

| Method | Recovery Rate (%) | | | | Macro F1 ($\times 100$) | | | |
|---|---|---|---|---|---|---|---|---|
| | Short | Medium | Long | All | Short | Medium | Long | All |
| SeqRNN(128) | 26.52±1.07 | 24.86±0.82 | 27.31±0.41 | 26.23±0.87 | 17.22±1.69 | 17.20±1.91 | 8.44±2.70 | 17.74±1.59 |
| SeqRNN(256) | 27.61±1.85 | 27.16±0.63 | 28.71±0.14 | 28.24±0.46 | 12.54±2.94 | 13.64±5.24 | 8.85±2.41 | 13.64±2.69 |
| SeqLSTM(128) | 23.48±1.07 | 26.32±0.05 | 26.78±1.12 | 24.70±0.64 | 9.89±0.57 | 10.44±1.42 | 10.71±2.53 | 10.28±0.61 |
| SeqLSTM(256) | 25.00±0.00 | 26.89±0.35 | 28.55±0.13 | 26.93±0.93 | 9.26±1.16 | 9.48±0.74 | 7.14±0.00 | 10.93±0.15 |
| StructMLP | 25.72±0.51 | 25.03±1.39 | 25.38±1.69 | 25.35±0.25 | 17.46±2.39 | 18.57±3.45 | 17.53±8.43 | 18.88±2.50 |
| StructGNN | 27.55±0.94 | 28.78±0.87 | 28.23±1.95 | 28.23±0.71 | 24.01±3.62 | 22.15±4.67 | 26.05±4.63 | 24.87±1.65 |
| GraphTrans | 26.15±0.93 | 23.78±1.11 | 23.80±1.69 | 24.73±0.93 | 16.34±2.67 | 16.39±4.74 | 18.67±7.16 | 17.18±3.81 |
| PiFold | 24.81±2.01 | 25.90±1.56 | 23.55±1.43 | 24.48±1.13 | 17.48±2.24 | 18.10±6.76 | 14.06±3.53 | 17.45±1.33 |
| RDesign | 37.22±1.14 | 44.89±1.67 | 43.06±0.08 | 41.53±0.38 | 38.25±3.06 | 40.41±1.27 | 41.48±0.91 | 40.89±0.49 |
| gRNAde | 42.40±1.41 | 45.10±0.92 | 43.42±1.62 | 43.26±0.71 | 39.22±0.59 | 45.22±1.23 | 42.21±1.30 | 43.09±0.39 |
| R3Design | 44.03±0.52 | 43.97±0.65 | 43.31±1.12 | 43.93±0.12 | 40.59±1.55 | 43.67±0.57 | 42.33±1.11 | 42.78±0.29 |
| Ours | **45.39±0.90** | **46.04±0.83** | **44.71±0.94** | **45.49±0.72** | **41.84±0.93** | **45.86±0.74** | **43.65±1.12** | **44.42±0.42** |

reported as mean ± standard deviation over three random seeds. To improve robustness, we apply structural perturbation during training by adding Gaussian noise to input atomic coordinates with $\sigma = 0.1$, which is consistent with typical coordinate fluctuations in experimentally determined or simulation-derived RNA structures.

## 4.2. Results on RDesign Split

The left part of Tab. 1 reports the recovery rate (%) of different models on the benchmark dataset across RNA sequences of varying lengths. Our ATL-FGD achieves 45.39±0.90, 46.04±0.83, and 44.71±0.94 recovery on short, medium, and long sequences, respectively, yielding an overall score of 45.49±0.72, the best among all compared methods. Compared with RDesign (Tan et al., 2024), which employs the same backbone with fixed graph topology but relies on contrastive representation learning, ATL-FGD consistently delivers higher recovery rates across all sequence length regimes. Specifically, our method surpasses RDesign by 8.17% on short sequences, 1.15% on medium sequences, and 1.65% on long sequences. Overall, ATL-FGD achieves a gain of 3.96% in the averaged recovery rate. When compared with gRNAde (Joshi et al., 2025), which incorporates multi-conformation learning, ATL-FGD still yields consistent improvements: 2.99%, 0.94%, and 1.29% on short, medium, and long sequences, respectively, leading to an overall improvement of 2.23%. The margin over R3Design (Tan et al., 2025), a refinement of RDesign by replacing MPNN layers with PiGNN (Gao et al., 2023) layers, is also notable at 1.56% in overall recovery. These results demonstrate that ATL-FGD effectively enhances structural representation learning through adaptive topology modeling and FiLM-guided distillation. The most pronounced improvement on short RNAs suggests that the proposed adaptive topology learning alleviates the noise introduced by fixed $k$NN graphs, which are particularly unstable when the number of nucleotides is small.

*Table 2.* The overall performance on gRNAde split. *Param.* denotes the parameter count (in millions).

| Method | Param. (M) | Recovery | Macro F1 |
|---|---|---|---|
| gRNAde | 2.15 | 42.15±2.20 | 45.96±1.92 |
| RDesign | 1.32 | 47.94±0.55 | 50.44±0.65 |
| R3Design | 3.23 | 48.53±0.66 | 52.21±0.57 |
| RiboDiffusion | 15.97 | 48.82±2.10 | 51.41±0.72 |
| Ours | 1.55 | 50.35±0.41 | 53.01±0.69 |

The right part of Tab. 1 further reports the Macro-F1 scores across RNA sequences of different lengths. Our ATL-FGD achieves 41.84±0.93, 45.86±0.74, and 43.65±1.12 on short, medium, and long sequences, respectively, resulting in an overall score of 44.42±0.42, the highest among all evaluated methods. Compared with RDesign, our ATL-FGD delivers a consistent improvement of 3.53 in overall score. While compared with gRNAde and R3Design, ATL-FGD further achieves gains of 1.33 and 1.64, respectively. Rather than offering marginal trade-offs between precision and recall, our model maintains a balanced performance across all regimes, indicating that our ATL-FGD effectively enhances the discriminative quality. Notably, the improvement in Macro-F1 aligns with the recovery rate trend but reflects a more stable enhancement in class-level per-nucleotide prediction balance, indicating that the proposed adaptive topology learning not only recovers correct nucleotides but also improves the overall prediction consistency of the generated sequences. In addition, beyond sequence-level metrics, we also report in the Appendix a comparison of structural fidelity between the designed and original RNAs.

## 4.3. Results on gRNAde Split and Efficiency

We further evaluate ATL-FGD on the gRNAde single-state split. For fair comparison, we reimplement all baselines under the same preprocessing and training protocol. Tab. 2 reports sequence recovery and Macro-F1, together with the number of parameters (in millions). ATL-FGD achieves

*Table 3.* Comparison of computational costs across different models, including training GPU memory consumption (GB), training epoch time (s), and evaluation epoch time (s).

| Model | Train Mem. | Train Time | Eval Time |
|---|---|---|---|
| MPNN basemodel | 3.87 | 4.00 | 4.10 |
| gRNAde | 6.48 | 4.60 | 5.14 |
| R3Design (w/o iter.) | 14.57 | 5.22 | 5.29 |
| R3Design (w/ iter.) | 42.13 | 8.20 | 8.80 |
| Ours (k=30) | 4.81 | 4.76 | 4.15 |
| Ours (k=64) | 8.93 | 5.68 | 4.22 |
| Ours (k=96) | 11.97 | 6.46 | 4.32 |

*Table 4.* The overall recovery rate and Macro-F1 scores on the Rfam and RNA-Puzzles datasets.

| Method | Recovery Rate | | Macro F1 | |
|---|---|---|---|---|
| | Rfam | RNA-Puz. | Rfam | RNA-Puz. |
| StructMLP | 24.40±1.63 | 24.22±1.28 | 16.79±4.01 | 16.40±3.28 |
| StructGNN | 27.64±3.31 | 27.96±3.08 | 23.84±3.45 | 22.51±4.15 |
| GraphTrans | 23.81±2.57 | 22.21±2.98 | 17.32±5.28 | 17.04±5.36 |
| PiFold | 22.55±4.13 | 23.78±6.52 | 16.08±2.34 | 16.32±4.62 |
| RDesign | 56.12±1.03 | 50.12±1.07 | 53.27±1.28 | 49.24±1.07 |
| gRNAde | 58.31±0.69 | 50.26±0.50 | 56.27±0.75 | 49.74±0.75 |
| R3Design | 62.77±1.42 | 54.34±0.42 | 60.64±1.29 | 54.99±0.56 |
| Ours | **64.33±0.36** | **55.34±0.85** | **62.24±0.56** | **56.05±1.23** |

the best overall performance while remaining compact, obtaining 50.35±0.41 recovery and 53.01±0.69 Macro-F1 with 1.55M parameters. In terms of computational footprint, ATL-FGD adds only a lightweight edge scoring function and two FiLM heads to the backbone. Against RDesign, which shares the same MPNN backbone, ATL-FGD achieves an improvement of 2.41% in recovery rate and 2.57 in Macro-F1, while keeping a comparable parameter budget. This gain highlights the complementary benefits of learning layer-adaptive topology via ATL and injecting foundation-model priors via FGD. Compared with R3Design, ATL-FGD improves recovery rate by 1.82% and Macro-F1 by 0.80, while using less than half the parameters of the AttnMLP-based backbone. Compared with the diffusion-based RiboDiffusion (Huang et al., 2024), ATL-FGD improves recovery rate by 1.53% and Macro-F1 by 1.60, while using a much smaller model. Notably, RiboDiffusion exhibits substantially higher variance across random seeds, e.g., STD of ±2.10 in recovery rate, suggesting that heavy diffusion-based backbone can be more sensitive to initialization and optimization settings in this task.

As shown in Tab. 3, we report the computational cost over 8 epochs and report the peak GPU memory usage observed during these runs. Increasing the candidate graph size introduces additional training cost for our method, but the overhead grows approximately linearly with $k$, and the evaluation-time cost remains marginal. In comparison, the iterative version of R3Design incurs a much larger increase in both memory consumption and latency, while yielding only a modest improvement. These results suggest that our method provides a practical computational trade-off.

### 4.4. Results on Generalization Datasets

To further evaluate cross-dataset generalization, we test ATL-FGD on two external benchmarks, Rfam and RNA-Puzzles, following the evaluation protocol of RDesign. As summarized in Tab. 4, ATL-FGD achieves a recovery rate of 64.33±0.36 and a Macro-F1 of 62.24±0.56 on Rfam, outperforming RDesign (Tan et al., 2024) by 8.21% and 8.97%, respectively. On RNA-Puzzles, ATL-FGD obtains

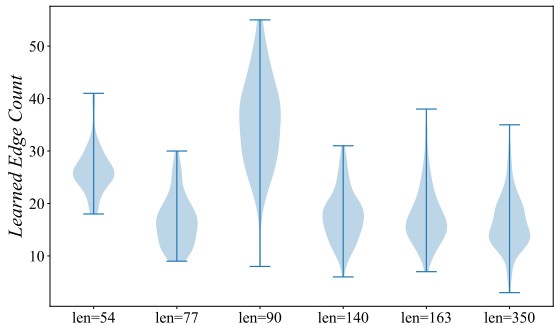

*Figure 3.* Learned edge count distribution over six samples with varying lengths.

55.34±0.85 recovery rate and 56.05±1.23 Macro-F1, outperforming RDesign by 5.22% and 6.81%, respectively. Compared with gRNAde(Joshi et al., 2025), which incorporates multi-conformation training, ATL-FGD still yields consistent gains of 6.02% and 5.97% on Rfam, while maintaining improvements of 5.08% and 6.31% on RNA-Puzzles. Although both benchmarks involve relatively small test sets, the overall margins remain clear and substantial. These results demonstrate that ATL-FGD generalizes reliably to unseen RNAs, validating that the combination of adaptive topology learning and distillation enhances the model's capacity to capture robust structural and contextual priors.

### 4.5. Visualization and Analysis

To better understand the behavior of our Adaptive Topology Learning, we visualize in Fig. 3 the learned edge count distribution at the last layer for six samples with varying sequence lengths. The violin plots show that the number of active edges learned by ATL exhibits substantial diversity across samples, rather than converging to a fixed pattern. To further illustrate how ATL refines nucleotide-level connectivity, we visualize the learned topology of an RNA sample (PDB ID 7DA7, length 90) in Fig.4. We display two nucleotides from distinct regions of the RNA, node #33 and node #89, and compare their neighborhoods under fixed

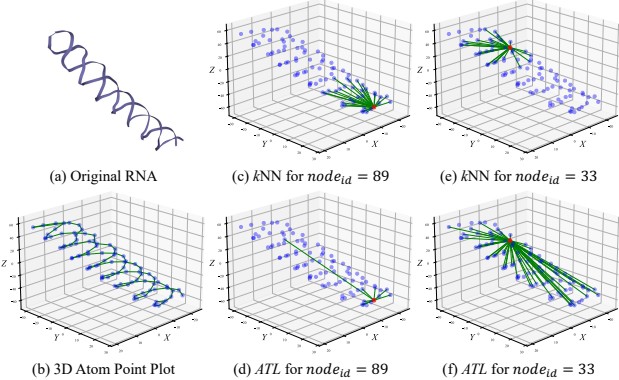

(a) Original RNA  (c) $k$NN for $node_{id} = 89$  (e) $k$NN for $node_{id} = 33$

(b) 3D Atom Point Plot  (d) $ATL$ for $node_{id} = 89$  (f) $ATL$ for $node_{id} = 33$

*Figure 4.* Visualization of the adaptive topology learned for RNA structure (PDB ID 7DA7). We highlight two representative nucleotides (node #33 and node #89) and visualize their learned connections in the final GNN layer compared with $k$NN.

$k$NN versus ATL. Node #33, located near the central region of the RNA, exhibits limited neighborhood coverage under fixed $k$NN which fails to capture long-range context-aware interactions. In contrast, ATL constructs a substantially richer and more global neighborhood, assigning edges to over half of the RNA, 55 active edges, suggesting that this node requires widespread context-aware interactions that a static graph cannot capture. In the opposite scenario, node #89 is positioned near the edge region of the RNA. Due to the fixed $k$ value, the graph forcibly connects node #89 to many distant and structurally irrelevant nucleotides, introducing substantial topological noise. ATL effectively suppresses these spurious edges, retaining only 7 meaningful local neighbors and one long-range interaction, resulting in a refined connectivity pattern with 8 total edges. Such heterogeneous edge distributions suggest that the model dynamically balances local cohesion and global communication, validating the effectiveness of ATL in learning context-aware and layer-dependent topologies. A more detailed analysis of the learned edge topology is provided in the Appendix, where we further illustrate how the adaptive connectivity evolves across layers.

## 4.6. Ablation Studies

**Analysis of the Effectiveness of Each Component.** To verify the contribution of each component, we conduct an ablation study by incrementally adding KD/FGD and ATL to the baseline model. As shown in Tab. 5, the baseline achieves a recovery rate of 41.54±0.57 and a Macro-F1 of 40.50±0.28. Introducing KD with vanilla feature alignment yields almost no improvement, suggesting that direct feature matching is insufficient under the modality gap and the mismatched contextual scope between sequence- and structure-based encoders. Replacing vanilla KD with FGD results in a clear gain of 2.14% in recovery rate. Notably, applying ATL alone improves the recovery rate to 44.70%,

*Table 5.* Effectiveness of proposed modules. *Rec.* denotes recovery rate for short.

| Settings | KD | ATL | Rec. | F1 |
|---|---|---|---|---|
| Baseline | ✗ | ✗ | 41.54±0.57 | 40.50±0.28 |
| w/o ATL | Vanilla | ✗ | 41.59±0.25 | 41.69±0.28 |
| w/o ATL | FGD | ✗ | 43.68±0.86 | 43.39±0.37 |
| w/o KD | ✗ | ✔ | 44.70±0.83 | 43.69±0.13 |
| Ours | FGD | ✔ | **45.49±0.72** | **44.42±0.42** |

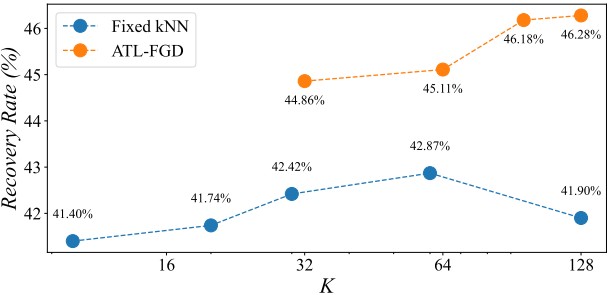

*Figure 5.* Recovery rate of fixed $k$NN graph construction and our ATL-FGD under different value of $k$.

indicating a larger benefit from adaptive topology learning. This suggests that the fixed $k$NN graph likely limits the message-passing context, while ATL mitigates this limitation by adapting the topology across layers. Finally, combining both modules, ATL-FGD achieves the best performance, with overall gains of 3.95% in recovery rate and 3.92% in Macro-F1 over the baseline, confirming that the proposed components are complementary and effective.

**Impact of Scaling Edge Count in Graph Modeling.** In this part, we investigate the impact of different value of $k$ in the $k$NN graph construction under different fashion. As shown in Fig. 5, the performance of the fixed-topology baseline exhibits a non-monotonic trend with respect to $k$. When $k$ increases from 10 to 60, the recovery rate gradually improves from 41.40% to 42.87%, indicating that moderate more global aggregation benefits structural representation. However, as $k$ continues to grow beyond 30, the performance saturates and even declines, reflecting the increasing influence of edge noise and irrelevant long-range connections introduced by static graph construction.

While introducing ATL, the model becomes significantly more robust to the choice of $k$. ATL-FGD achieves 44.86% with $k = 32$, outperforming the fixed topology by 2.44%, validating that ATL effectively filters out spurious edges while preserving critical ones. Moreover, the performance continues to improve as $k$ increases, achieving 46.28% at $k = 128$. This trend aligns with our motivation that ATL leverages large $k$ as a redundant candidate graph and dynam-

*Table 6.* The recovery rate with different value of $\tau$.

| $\tau$ | Short | Medium | Long | All |
|------|-------|--------|------|-----|
| 0.3 | 45.52 | 45.36 | 45.13 | 45.44 |
| 0.4 | **45.61** | 45.87 | 45.30 | **45.65** |
| 0.5 | 45.25 | 45.91 | **45.73** | 45.48 |
| 0.6 | 44.87 | **46.05** | 45.70 | 45.29 |
| 0.7 | 44.88 | 45.35 | 45.16 | 45.04 |

*Table 7.* Performance of ATL under larger candidate graphs.

| Setting | Short | Medium | Long | All |
|---------|-------|--------|------|-----|
| $k = 96$ | 45.76±1.38 | 46.32±2.72 | 44.09±0.96 | 45.72±0.86 |
| $k = 256$ | 45.32±0.98 | 43.89±0.89 | 44.26±0.20 | 44.80±0.85 |
| $k = 512$ | 45.27±1.73 | 44.35±1.28 | 43.77±0.67 | 44.83±1.36 |

ically learns context-aware connectivity, while suppressing noise from irrelevant neighbors. These observations along with the phenomenon displayed in Fig. 3 and Fig. 4 further confirm that ATL could alleviate the issues of fixed graph construction in a learnable fashion.

**Impact of Threshold $\tau$.** We further investigate the sensitivity of ATL to the edge selection threshold $\tau$, which controls the sparsity of the learned topology during inference. Specifically, we fix the candidate graph size as $k = 96$ and vary $\tau$ from 0.3 to 0.7. As shown in Tab. 6, the overall recovery remains stable across different thresholds, ranging only from 45.04% to 45.65%. This indicates that ATL is not sensitive to delicate threshold tuning. Across different RNA length groups, the performance also shows only minor fluctuations. A larger threshold imposes stronger pruning on candidate edges, which can help remove more connections and slightly improve performance in some cases. However, overly aggressive pruning may also discard useful structural interactions, leading to a mild performance decrease. Therefore, we use $\tau = 0.5$ as the default setting, as it provides a stable middle-ground choice.

**Behavior under Overly Dense Candidate Graphs.** We further examine the quantitative behavior of ATL when the candidate graph is pushed to much denser regimes under the ATL-only setting. As shown in Tab. 7, increasing the candidate graph from $k = 96$ to denser settings does not further improve the overall recovery. Instead, the performance slightly decreases. A sufficiently large candidate graph provides ATL with a broader structural search space, allowing the model to select informative edges adaptively. However, an excessively dense graph also introduces many redundant or weakly informative edges, which enlarges the gating search space and may dilute useful structural signals. Although some additional edges can be admissible according to distance-based graph construction, their geometric cues, such as orientation, distance, and direction, may still be ambiguous or weakly related to functional structural interactions. Moreover, long RNAs are relatively scarce in the dataset, which may further limit the model's ability to learn reliable adaptive edge selection under extremely large $k$.

## 5. Conclusion

In this work, we present ATL-FGD, a unified framework for tertiary structure–based RNA design that advances both structural representation learning and cross-modal knowledge transfer. Our Adaptive Topology Learning (ATL) replaces static $k$NN graph connectivity with layer-wise, data-dependent topologies, enabling context-aware message passing that better captures heterogeneous tertiary interactions. In parallel, FiLM-Guided Distillation (FGD) bridges structure- and sequence-based representations by transferring RNA foundation model priors to the structural encoder via feature-wise modulation. Across multiple benchmarks, ATL-FGD yields consistent improvements in recovery rate and Macro-F1, and generalizes robustly to datasets such as Rfam and RNA-Puzzles. These results underscore the value of adaptive graph modeling and principled representation transfer for structure-informed RNA design.

In future work, we plan to extend ATL-FGD toward joint RNA–protein co-design and broader multi-modal generative modeling, further integrating geometric reasoning with large-scale biochemical knowledge to support structure-guided molecular engineering.

## Acknowledgements

This work was supported in part by Guangdong S&T Programme with Grant No. 2024B0101030002, the Basic Research Project No. HZQB-KCZYZ-2021067 of Hetao Shenzhen-HK S&T Cooperation Zone, the Shenzhen Outstanding Talents Training Fund 202002, the NSFC with Grant No. 62293482, by NSFC with Grant No. 62573371, by the Guangdong Province Radio Science Data Center with grant No. 2025B1212070001, by the Shenzhen General Program No. JCYJ20220530143600001, by the Guangdong Research Project No.2017ZT07X152 and No. 2019CX01X104, by the Guangdong Provincial Key Laboratory of Future Networks of Intelligence (Grant No. 2022B1212010001), by the Guangdong Provincial Key Laboratory of BigData Computing CHUK-Shenzhen, by the NSFC 61931024&12326610, by the Shenzhen Key Laboratory of Big Data and Artificial Intelligence (Grant No. SYSPG20241211173853027), by National Key Research and Development Program of China: 2025YFF0515300 and 2025YFF0515304, by China Association for Science and Technology Youth Care Program, the Open Project Program (Grant No. QHSF-CS-2606) of Key Laboratory of Tibetan Information Processing, Ministry of Ed-

ucation, by the Shenzhen-Hong Kong Joint Funding No. SGDX20211123112401002, and by Tencent & Huawei Open Fund.

## Impact Statement

The goal of our paper is to advance computational RNA structure-aware design via deep learning. More accurate and data-efficient computational design may support research in RNA therapeutics and related areas by accelerating early-stage in silico screening and hypothesis generation, with potential downstream relevance to precision medicine. At the same time, our method relies on large-scale biological datasets that may exhibit limited coverage, selection bias, and measurement noise due to constraints in experimental structure determination. In addition, our method is evaluated only on public benchmarks and has not been validated in wet-lab settings, experimental verification remains essential before any real-world use. Finally, as with many advances in biomolecular design, there is a potential for dual-use if such methods are applied to engineer harmful biological functions. We focus on the computational modeling aspect and do not describe wet-lab procedures, any downstream experimental use should follow standard biosafety and regulatory requirements.

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

## A. Qualitative Analysis of Learned Topology

To further analyze how ATL evolves across the network depth, we visualize in Fig. 6 the layer-wise learned edge count distributions for six representative samples. Consistent with the sample-level diversity observed in the main text, the violin plots reveal that each GNN layer learns a distinct connectivity pattern, rather than converging to a uniform or monotonically increasing topology. Early layers generally preserve broader neighborhoods with higher edge counts, enabling coarse-grained aggregation, while intermediate layers often become more selective as the model refines local structural cues. In several cases (sample ID 140 and 190), the final layer again expands its connectivity, suggesting that ATL dynamically balances local refinement and global context integration throughout the hierarchy. This layer-dependent heterogeneity highlights that adaptive topology learning is not only sample-specific, but also layer-wise adaptive, allowing the model to adjust information flow at different stages of structural reasoning.

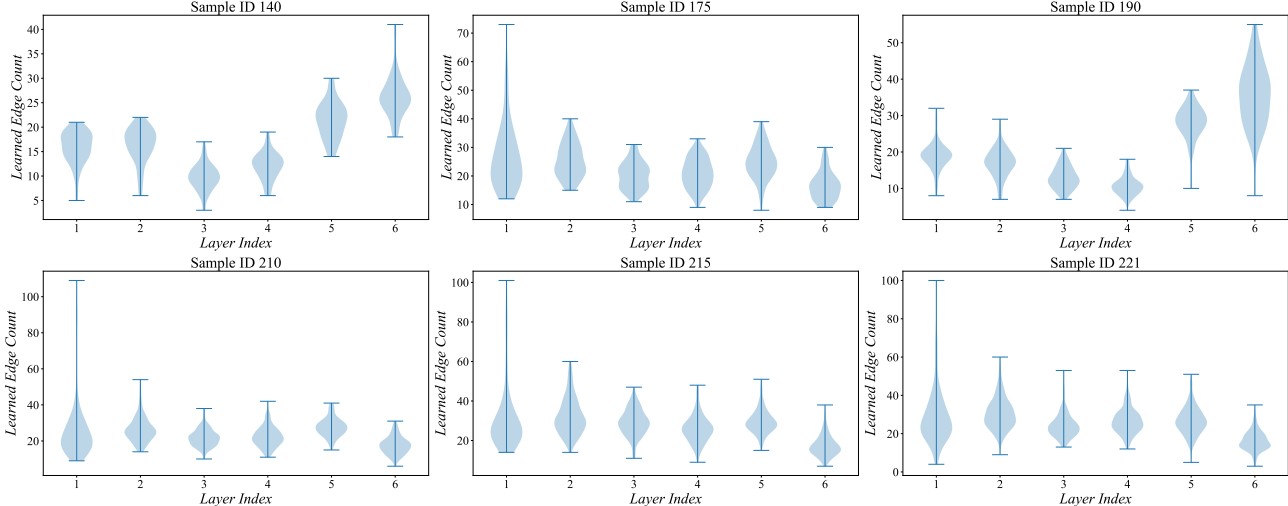

*Figure 6.* Learned edge count distribution of 6 samples across layer.

To further investigate how ATL adapts connectivity across network depth, we visualize the layer-wise adjacency matrices for sample ID 140 and sample ID 190 in Fig. 7 and Fig. 8. As shown in Fig. 9, both RNAs exhibit helical structure, and this characteristic is reflected in the learned topology that in the early GNN layer, ATL retains an X-shaped connectivity pattern that mirrors the local geometric arrangement of the helix. As message passing proceeds through deeper layers, the model progressively expands and enriches its connectivity, preserving additional long-range interactions that are essential for maintaining the global structural organization. This progression illustrates that ATL does not simply prune edges, instead, it adaptively enhances structure-specific relationships as layers aggregate more contextual information, reinforcing the underlying topology of each sample.

We further analyze the learned topologies at the dataset level by grouping test RNAs into three length buckets. As summarized in Tab 8, ATL learns sparse and layer-dependent subgraphs from the dense candidate graph. Notably, for short and medium RNAs with lengths no greater than 100 nucleotides, the $k$NN candidate graph is almost fully connected because $k$ exceeds

*Table 8.* Statistics of learned topologies under the $k = 128$ setting. For each group, we report the number of samples, the mean retained degree, and the mean retained edge ratio across different layers.

| Length | Metric | #Samples | Layer 1 | Layer 2 | Layer 3 | Layer 4 | Layer 5 | Layer 6 |
|--------|--------|----------|---------|---------|---------|---------|---------|---------|
| S | Mean degree | 136 | 10.24±5.10 | 11.63±5.53 | 10.08±4.38 | 11.33±5.49 | 12.23±6.00 | 10.69±5.23 |
| S | Mean ratio | 136 | 0.55±0.19 | 0.62±0.18 | 0.57±0.21 | 0.62±0.20 | 0.65±0.17 | 0.59±0.20 |
| M | Mean degree | 61 | 21.22±4.13 | 21.94±2.54 | 17.77±2.59 | 19.07±4.65 | 22.01±2.53 | 16.63±3.69 |
| M | Mean ratio | 61 | 0.30±0.06 | 0.31±0.05 | 0.25±0.05 | 0.27±0.09 | 0.31±0.04 | 0.24±0.06 |
| L | Mean degree | 26 | 26.17±2.80 | 27.02±2.84 | 21.82±3.02 | 22.81±3.16 | 26.85±2.03 | 16.83±1.81 |
| L | Mean ratio | 26 | 0.21±0.02 | 0.22±0.02 | 0.18±0.03 | 0.19±0.03 | 0.22±0.02 | 0.14±0.02 |

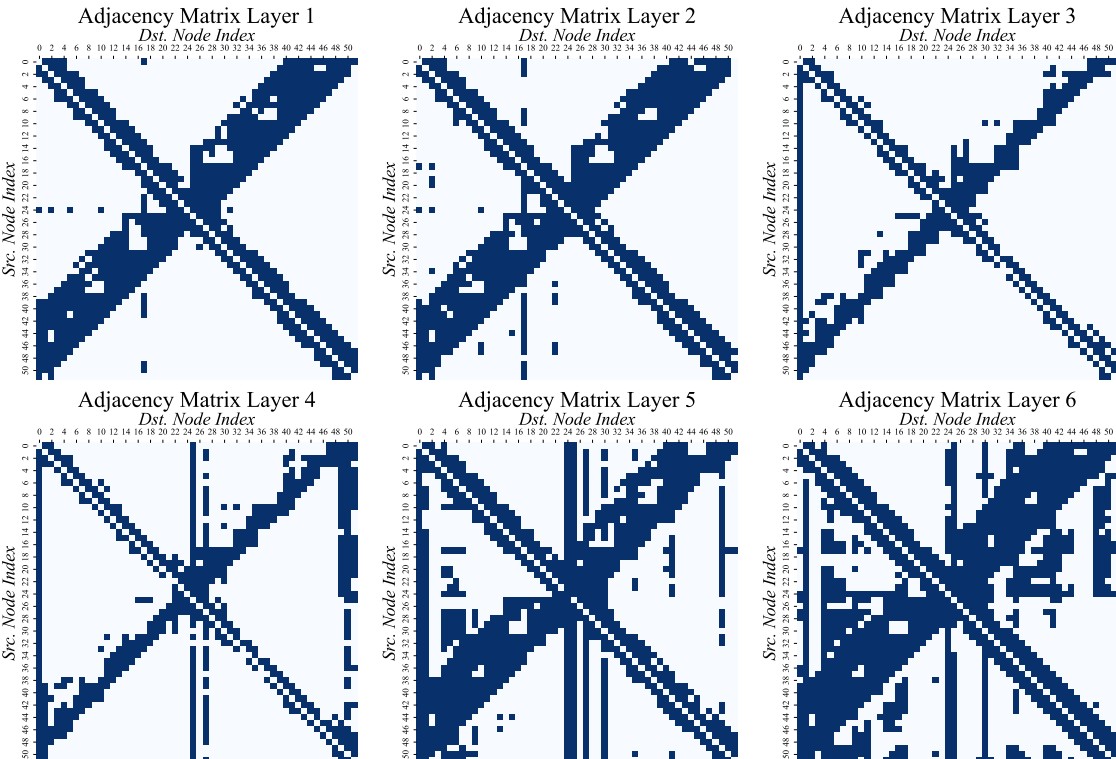

*Figure 7.* Detailed graph topology across layer for sample ID 140 (7XW2).

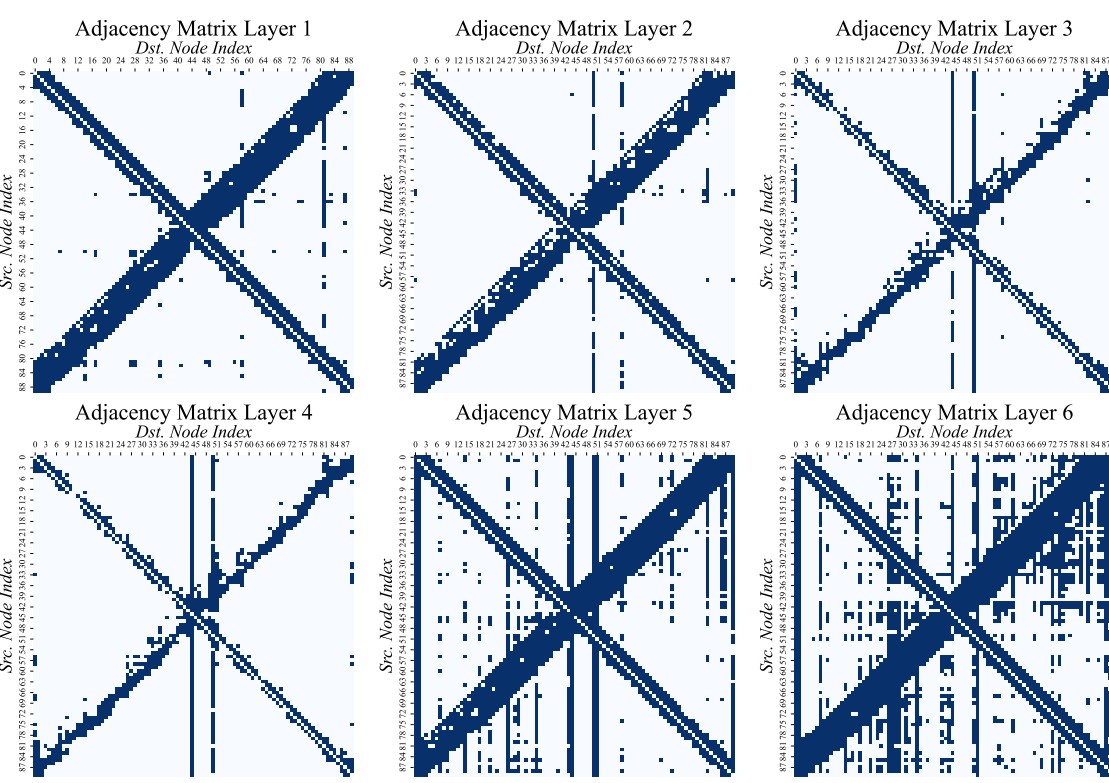

*Figure 8.* Detailed graph topology across layer for sample ID 190 (7DA7).

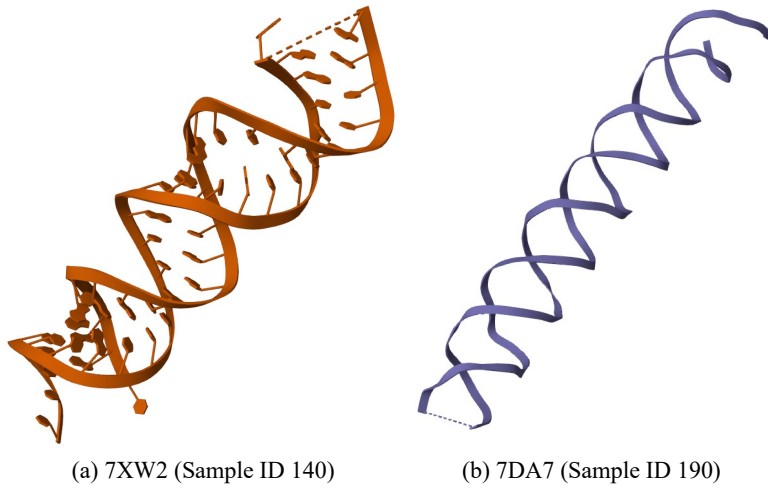

(a) 7XW2 (Sample ID 140)      (b) 7DA7 (Sample ID 190)

*Figure 9.* Original RNA structures of sample ID 140 and 190.

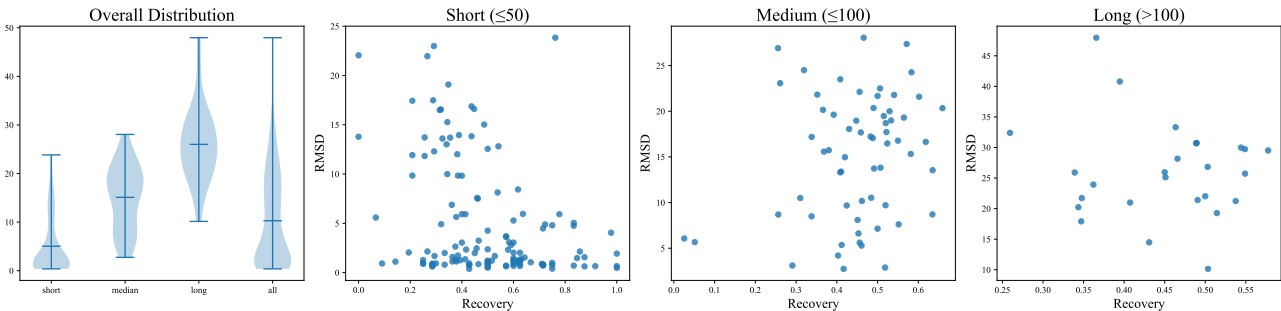

*Figure 10.* Structural consistency analysis of designed RNA sequences.

or approaches the sequence length. Even under this dense candidate setting, ATL retains only 55% ∼ 65% of candidate edges for short RNAs and further reduces the retained ratio to 24% ∼ 31% for medium RNAs. For long RNAs, the retained ratio is even lower, ranging from 14% to 22% across layers. Moreover, the retained connectivity changes non-monotonically across layers in all three length groups, indicating that ATL does not simply apply a fixed pruning rule but instead learns layer-dependent adaptive subgraphs.

## B. Structural Fidelity Evaluation

To assess the structural fidelity of the designed RNA sequences of the test set of the RDesign split, we fold all predicted sequences using Protenix and compute the backbone RMSD between the predicted structures and the reference tertiary structures. Fig. 10 summarizes the distribution of RMSD across length regimes, including a global violin plot and scatter plots for short, medium, and long sequences. Across all the samples, our designs achieve an average RMSD of 10.28 with a standard deviation of 9.61, while the corresponding TM-score averages 0.426 with a standard deviation of 0.211, indicating reasonable global structural similarity. While for samples grouped by sequence length, short sequences achieve a mean RMSD of 5.04 with a STD of 5.88, medium sequences exhibit a mean RMSD of 15.10, and long sequences achieve a higher RMSD of 26.01. This progressive increase in RMSD aligns with the expected rise in tertiary structural complexity as RNA length grows, reflecting the greater structural flexibility and folding ambiguity inherent in RNA.

