# OpenReview forum: "Learning Adaptive Topology with FiLM-Guided Distillation for Tertiary Structure-Based RNA Design"
_ICML.cc/2026/Conference — ICML 2026 regular_

### Official Review · Reviewer_X5NH · 2026-03-09

**Soundness:** 3
**Presentation:** 2
**Significance:** 3
**Originality:** 3
**Overall Recommendation:** 4
**Confidence:** 4

**Summary:**

Leveraging Feature-wise Linear Modulation (FiLM), this paper studies tertiary structure-based RNA design and proposes ATL-FGD, a framework that combines Adaptive Topology Learning with FiLM-Guided Distillation. The main idea is to replace static graph connectivity with layer-wise learnable edge gating, so that each GNN layer operates on a dynamically selected subgraph rather than on a fixed kNN graph. The paper also introduces a distillation pathway from a frozen RNA foundation model, RiNALMo 650M, where the student is trained to match teacher-induced FiLM parameters for knowledge distillation.

**Compliance With Llm Reviewing Policy:**

Affirmed.

**Final Justification:**

The rebuttal has addressed main concerns. The evaluation score has been changed to the positive side.

**Key Questions For Authors:**

- Can the authors clarify the exact training dynamics of the FiLM-guided distillation module?
- Could the authors provide quantitative statistics describing the learned topology after adaptive gating?
- In the k-sensitivity experiment, performance continues to improve as the candidate neighborhood size increases up to $k=128$. Could the authors comment on whether further increasing k, or even using a fully connected candidate graph, would continue to benefit ATL?

**Limitations:**

yes

**Strengths And Weaknesses:**

### Strengths
The paper addresses a problem in tertiary structure guided RNA sequence design: how to capture the flexible and heterogeneous nature of RNA structures. The proposed ATL module introduces a simple mechanism to adapt graph connectivity through learnable edge gating while retaining geometric priors. Experiments show consistent improvements across several benchmarks, and the ablation study provides some evidence that the adaptive topology component contributes to the performance gains.

### Weaknesses
- The paper does not clearly describe how the FiLM heads are initialized or updated during training. If the teacher FiLM head is optimized jointly with the student network, the FiLM parameters used as distillation targets may also change during training. This makes it less clear whether the student is truly distilling information from the teacher, or whether the teacher FiLM head is adapting to reduce the distillation loss. In addition, the paper does not clearly explain how the two student representations $\hat{h}_{stu}$ and $\tilde{h}_{stu}$ are used in the task loss.
- The current ablation study does not rule out an alternative explanation that the performance gain mainly comes from the additional student FiLM modulation capacity rather than from teacher knowledge transfer. A useful control experiment would be to keep the student FiLM head while removing teacher-guided distillation.
- The paper attributes ATL’s improvement to reducing noise from fixed kNN graphs, but the analysis of the learned topology remains limited. Figure 3 illustrates variability in active edges but does not sufficiently characterize the learned graph structure. Statistics such as average active degree, edge pruning ratio, or edge-length distribution would provide stronger evidence.

---

> ### Author Rebuttal · Authors · 2026-03-30
>
> We thank the reviewer for the careful reading and the helpful suggestions.
>
> ### 1. On the FiLM-guided distillation dynamics
>
> During training, the teacher encoder is frozen, and the teacher/student FiLM heads generate two sets of modulation parameters $(\gamma,\beta)$ from $h_{tea}$ and $h_{stu}$, respectively.
> Both modulated student representations are passed through the same readout head:
> $pred_1 = readout(\hat h_{stu}), pred_2 = readout(\tilde h_{stu}),$
> and the task loss is $L_{task} = CE(pred_1) + CE(pred_2)$.
> During training, only the params of FiLM module in teacher branch are optimized.
>
> ### 2. Student FiLM capacity alone
>
> To address the reviewer’s suggested control, we additionally evaluated a student-FiLM-only variant that keeps the student FiLM head while removing teacher-guided distillation.
> This variant achieves 42.07$\pm$0.59.
> This result indicates that extra modulation capacity is indeed partially helpful, but it does not explain the full gain of FGD, which remains substantially stronger.
> For more detailed analysis about KD, please refer to the rebuttal of Sec. 3 for Reviewer yqSg.
>
> ### 3. Quantitative behavior of ATL with larger candidate graphs
>
> Regarding larger candidate graphs, we further tested ATL-only under much denser candidate graphs (averaged over 3 runs):
>
> | Setting | Short | Medium | Long | All |
> |:-:|:-:|:-:|:-:|:-:|
> | k=96 | 45.76 | 46.32 | 44.09 | 45.72 |
> | k=256 | 45.32 | 43.89 | 44.26 | 44.80 |
> | k=max_length | 45.27 | 44.35 | 43.77 | 44.83 |
>
> This shows that enlarging the candidate graph is helpful up to a point: ATL benefits from a sufficiently expressive candidate graph, but an excessively dense graph introduces redundant edges, enlarges the gating search space, and may dilute useful structural signals.
> Since edge features are composed of orientation, distance, and direction, some added edges may be admissible by distance yet still provide weak or ambiguous geometric cues.
> In addition, long RNAs (>100 nts) are relatively scarce in the dataset (about 10%), so learning reliable adaptive selection under extremely large k may also be data-limited.
>
> ### 4. Additional statistics for learned graph
>
> | Length | #Samples | Layer 1 | Layer 2 | Layer 3 | Layer 4 | Layer 5 | Layer 6 |
> |:-:|:-:|:-:|:-:|:-:|:-:|:-:|:-:|
> | S (mean deg) | 136 | 10.24$\pm$5.10 | 11.63$\pm$5.53 | 10.08$\pm$4.38 | 11.33$\pm$5.49 | 12.23$\pm$6.00 | 10.69$\pm$5.23 |
> | S (mean ratio) | 136 | 0.55$\pm$0.19 | 0.62$\pm$0.18 | 0.57$\pm$0.21 | 0.62$\pm$0.20 | 0.65$\pm$0.17 | 0.59$\pm$0.20 |
> | M (mean deg) | 61 | 21.22$\pm$4.13 | 21.94$\pm$2.54 | 17.77$\pm$2.59 | 19.07$\pm$4.65 | 22.01$\pm$2.53 | 16.63$\pm$3.69 |
> | M (mean ratio) | 61 | 0.30$\pm$0.06 | 0.31$\pm$0.05 | 0.25$\pm$0.05 | 0.27$\pm$0.09 | 0.31$\pm$0.04 | 0.24$\pm$0.06 |
> | L (mean deg) | 26 | 26.17$\pm$2.80 | 27.02$\pm$2.84 | 21.82$\pm$3.02 | 22.81$\pm$3.16 | 26.85$\pm$2.03 | 16.83$\pm$1.81 |
> | L (mean ratio) | 26 | 0.21$\pm$0.02 | 0.22$\pm$0.02 | 0.18$\pm$0.03 | 0.19$\pm$0.03 | 0.22$\pm$0.02 | 0.14$\pm$0.02 |
>
> We further summarize the dataset level learned topology under k=128 by grouping test RNAs into three length buckets.
> We will include more detailed statistics in the appendix in the revision.
> Notably, for short and medium RNAs (<=100 nts), the candidate graph by kNN is fully connected.
> For short RNAs, ATL retains only 55%-65% of candidate edges across layers; for medium RNAs, the retained ratio further decreases to 24%-31%.
> Moreover, the retained connectivity changes non-monotonically across layers in all three groups, indicating that ATL learns layer-dependent subgraphs.

---

> > ### Author Rebuttal · Reviewer_X5NH · 2026-04-04
> >
> > The rebuttal clarifies that the primary improvement comes from distillation rather than from adding the FiLM module.

---

> > > ### Author Response · Authors · 2026-04-05
> > >
> > > We sincerely thank the reviewer for the follow-up and greatly appreciate the positive acknowledgement that our rebuttal has addressed the concerns.
> > >
> > > As clarified in the rebuttal, an important point is that the gain of FGD comes from teacher-guided transfer rather than simply from adding extra FiLM capacity.
> > > At the same time, we would also like to emphasize that the contribution of our paper is not limited to the distillation component alone.
> > > Our ATL-FGD is motivated by two needs: first, to move beyond static kNN graphs by learning adaptive, layer-wise connectivity that can suppress noisy or irrelevant edges; second, to improve representation learning under data scarcity by transferring useful sequence-level priors through distillation.
> > > From this perspective, ATL is intended to learn a more suitable layer-wise working subgraph rather than merely tune graph density, and our additional rebuttal results were included to make this point clearer.
> > > Likewise, FGD is designed as a FiLM-guided transfer of sequence priors that is better suited to the structure/sequence gap than direct feature alignment.
> > > In particular, the rebuttal helps clarify that the gain from FGD comes from teacher-guided knowledge transfer rather than simply adding extra FiLM capacity.
> > > In the rebuttal, we further added control analyses and clarifications to make these more explicit, including evidence that the improvement cannot be reduced to additional FiLM capacity alone and that ATL contributes beyond generic graph-density adjustment.
> > >
> > > More broadly, we believe this interpretation is also consistent with the feedback from other reviewers.
> > > For example, Reviewer Ndxf explicitly noted that the paper "accurately grasps the two practical bottlenecks of fixed topology and scarcity of RNA 3D data" and that ATL "learns the working subgraph layer by layer rather than simply tuning k."
> > > Reviewer sDMg highlighted that using the RNA foundation model only as a teacher "makes the comparison setting cleaner and the reported gains more convincing," while also appreciating that ATL is a "lightweight mechanism" that can be integrated into existing GNN backbones.
> > > Reviewer yqSg further emphasized that the experimental results are "sound" and that ATL-FGD effectively improves robustness to kNN sensitivity, supported by ablation studies.
> > > Together, these observations align with our original motivation and make the respective roles of ATL and FGD substantially clearer.
> > >
> > > Since your concerns have now been addressed, we would be very grateful if you could kindly reconsider the current score in light of the clarification and additional evidence provided in the rebuttal.

---

### Official Review · Reviewer_yqSg · 2026-03-12

**Soundness:** 3
**Presentation:** 3
**Significance:** 3
**Originality:** 3
**Overall Recommendation:** 5
**Confidence:** 5

**Summary:**

The paper proposes ATL-FGD, a framework for RNA inverse folding. By jointly integrating ATL - which replaces fixed kNN graph construction with a learnable layer wise module, and FGD - which performs distillation from frozen RNA foundation model via FiLM modulation parameters, it shows meaningful gains on top of existing benchmark datasets.

**Compliance With Llm Reviewing Policy:**

Affirmed.

**Final Justification:**

Most of my concerns are resolved, and have thus raised my score further leaning towards a stronger acceptance.

**Key Questions For Authors:**

1. Relation with oversmoothing. The papers’ main claim seems to be: large k→ biologically irrelvant edges → noise in message passing →degraded performance. However, I think a more plausible argument would be to relate the concept to oversmoothing - the performance degradation being caused by the oversmoothing of node representations. Can the authors provide evidence that oversmoothing is a different problem than the ones they described? If oversmoothing is indeed the cause, existing graph sparsification techniques can be used (e.g. DropEdge) as mitigation, degrading the proposed contribution of the authors.
2. Questions on adaptive topology mechanism. What prevents the ATL from collapsing? STE has been known to be sensitive to the choice of hyperparameters, and can easily fail to learn effectively. How does the ATL avoid this well-known problem? Can the authors provide additional ablations plotting during training / evaluation, where it plots the variance and mean of edges to give guidance as to whether the adaptive topology is indeed distinct across nodes, across training steps?
3. Is the improvement from FGD attributed to the FiLM mechanism or the foundation model distillation in general? Although the ablation is done upon vanilla KD, this is weak baseline for distillation. There are multiple works (most recently, REPA, relational distillation, contrastive distillation, etc.) that address the problem of injecting knowledge into models. Despite these methods not having been discussed widely in the domain of the paper, it should be compared against the FGD, for most of its mechanisms work in a plug-and-play style manner on top of existing models. Can the authors provide a comparison by selecting one of the recent distillation methods (preferrably REPA) and integrating it with one of the baseline models to give a comparison on performance with their proposed method?

With these questions adequately resolved, I am willing to raise my score.

**Limitations:**

Refer to the questions above.

**Strengths And Weaknesses:**

### Strengths
* Overall, the experimental analysis is sound. The authors provide multiple results that highlight the efficacy and generalizability of their model. The ablation results provided are convincing - where each component shows meaningful gains in performance.
* Robustness to the choice of k is one of the most highlighted factor. kNN based methods show degenerated performance with high k values (indicating hyperparameter sensitivity) whereas the ATL-FGD sidesteps this.
* Paper itself is generally well-organized with clear problem statements and logical flow. Visualization upon the adjacency matrix that adapt via node and layer is critical in their presentations, highlighting that ATL-FGD indeed learns to attend differently.
* The method itself seems novel, with their designs thoroughly motivated in their introduction. It is adequately designed for the RNA-modality specific characteristics.

### Weakness
* The ATL and FGD are orthogonal rather than additive. Despite the performance gains of both components, the method itself seems orthogonal. The two components aren't creating a joint synergy in performance, nor do they interact with one another to compensate each others' potential flaws.
* Novelty of FGD. There are multiple recent distillation methods, which are not compared against FGD. Authors do not provide further empirical reasoning / analysis on whether the FiLM based strategy is indeed better in the RNA modality.
* Novelty of demonstrated flaw in kNN. There are works discussing on graph oversmoothing - which the authors do not provide connections to. The reader might & will be confused on whether the kNN flaw stems from graph oversmoothing or the authors' proposal on "noise". This needs further empirical/theoretical clarification.

---

> ### Author Rebuttal · Authors · 2026-03-30
>
> We thank the reviewer for the careful reading and constructive comments.
>
> ### 1. On over-smoothing vs. noisy / irrelevant edges
>
> To directly examine whether the gain mainly comes from generic edge sparsification or alleviating over-smoothing, we conducted an additional control experiment using ATL only (without distillation) and compared it against DropEdge under the same backbone:
>
> | Method | k=64 | k=96 |
> |:-:|:-:|:-:|
> | baseline | 41.76$\pm$0.92 | 41.70$\pm$0.70 |
> | DropEdge | 42.77$\pm$0.33 | 43.44$\pm$1.06 |
> | ATL | **44.21$\pm$0.83** | **45.72$\pm$0.86** |
>
> These results suggest that generic sparsification does help, so over-smoothing / noisy message passing is indeed a relevant factor. However, ATL remains substantially stronger than DropEdge, especially at larger candidate neighborhoods, indicating that the degradation of dense fixed kNN graphs cannot be fully explained by over-smoothing alone. In our setting, irrelevant edges and suboptimal message routes also matter, and ATL provides a data-dependent way to refine the working graph beyond random edge regularization.
>
> We also observe that DropEdge performs particularly poorly on the medium and long subsets (41.65% and 42.31% under \(k=64\)), whereas ATL achieves 45.42% and 44.20%, respectively. This further suggests that ATL is not merely reducing connectivity, but is learning a more effective layer-wise subgraph than random sparsification.
>
> ### 2. On whether ATL collapses under STE-based gating
>
> STE-based binary gating can in principle be unstable and may collapse to trivial all-on/all-off patterns. In our implementation, the training-time gate follows a score -> probability (sigmoid) -> hard mask pipeline. The STE mainly ensures that the discrete gate remains trainable,
> $m = m_{\text{hard}} + (p - p.\mathrm{detach}())$,
> so the forward pass uses hard edge selection, while gradients are propagated through the underlying continuous probabilities p and the scoring head throughout training.
>
> In addition, the effective number of sampled gates per batch is large during training. With batch size 64 and k=96, a batch contains on average **52.0 nodes** and **3680.3 edges**; across **6 layers**, this corresponds to roughly **2.2$\times$10^4** gate variables per batch.
> We conjecture that although each edge gate is sampled stochastically, the Bernoulli noise is substantially averaged out at the batch/layer level, which could make the training dynamics relatively stable in practice.
>
> Empirically, we do not observe phenomenons of collapse. ATL remains relatively stable across different candidate graph sizes (from 32 to 256), and the learned topology shows non-trivial variation across layers and samples rather than degenerate uniform patterns.
> We will add more quantitative gate statistics (e.g., mean/variance of retained edges across node or during training) in the appendix in the revision to make this clearer.
>
> ### 3. Additional statistics for learned graph
>
> We also quantified the learned graph statistics (please refer to the Sec. 4 of rebuttal to X5NH for details), which show clear length-dependent and layer-dependent sparsity patterns rather than a trivial fixed pruning behavior.
> Due to the rebuttal space limit, we will include more comprehensive analyses in the revision appendix, including finer-grained node-level statistics and the evolution of the learned topology over training.
>
> ### 4. On whether FGD helps because of FiLM or because of distillation in general
>
> To strengthen this point beyond vanilla KD, we additionally compared FGD against several stronger distillation baselines under the same backbone:
>
> | Method | Recovery |
> |:-:|:-:|
> | REPA | 41.85$\pm$0.59 |
> | CKD | 42.04$\pm$0.94 |
> | RKD | 42.27$\pm$1.20 |
>
> REPA remains fundamentally a representation-level alignment method between student intermediate features and teacher representations. This direct positional/feature-level alignment is suboptimal because the RNA language model (RLM) teacher is transformer-based and captures global-context representations from sequence data, whereas the MPNN student is driven by local graph message passing from structure data. This teacher-student mismatch makes naive position-level alignment, as in REPA/CKD, less suitable.
>
> RKD performs slightly better than REPA/CKD, which is consistent with this interpretation, since RKD aligns relative relations/distances rather than relying purely on token-wise alignment. We also observe from t-SNE visualization that RLM and MPNN features exhibit different clustering structures, suggesting that direct feature alignment is not ideal in this cross-modal setting.

---

> > ### Author Rebuttal · Reviewer_yqSg · 2026-04-01
> >
> > Most of my concerns have been well-addressed. I will accordingly raise my score, leaning towards acceptance.

---

### Official Review · Reviewer_Ndxf · 2026-03-13

**Soundness:** 3
**Presentation:** 2
**Significance:** 3
**Originality:** 3
**Overall Recommendation:** 4
**Confidence:** 4

**Summary:**

In this paper, inverse folding driven by RNA tertiary structure is studied. ATL-FGD is proposed. On the one hand, Adaptive Topology Learning (ATL) is used to do layer-by-layer gate control on the candidate structure diagram to reduce noise caused by fixed kNN graphs; On the other hand, the prior of the frozen RNA foundation model is distilled into the structure encoder by FiLM-Guided Distillation (FGD). The overall framework is still based on the standard paradigm of "structural diagram coding+point-by-point sequence prediction", and adopts the same 6-layer MPNN backbone as RDesign. The experiment covers RDesign split, gRNAde split, Rfam, RNA-Puzzles, and includes ablation and K-value sensitivity analysis.

**Compliance With Llm Reviewing Policy:**

Affirmed.

**Final Justification:**

I appreciate the additional experiments and clarifications. These responses address several of my concerns and improve my confidence in the work.

**Key Questions For Authors:**

1) Can you provide direct structural-fidelity comparisons against the main baselines under the same folding/evaluation protocol?

2) Is ATL essentially just screening the candidate graph? How sensitive is performance to missing long-range contacts in the initial graph?

3) Does removing iterative generation from R3Design affect the fairness of the comparison? Could you also report results under the original setting?

**Limitations:**

yes

**Strengths And Weaknesses:**

Strengths

The problem is important, and the motivation of the method is clear: the author accurately grasps the two practical bottlenecks of fixed topology and scarcity of RNA 3D data.

The method has some new ideas: ATL is not simply tuning k, but learning the working subgraph layer by layer; FGD is not a regular feature matching, but an alignment FiLM parameter.

The whole experiment is solid: compared with many existing methods, the results are given in two main split, two external generalization sets, module ablation and K sensitivity. ATL-FGD is superior to strong baselines such as RDesign/gRNAde/R3Design in both main table and generalization set.

Weaknesses

The strongest claim is not supported enough. The paper claims to improve structural fidelity in many places, but the main experiment in the text mainly reports recovery/Macro-F1；Structural fidelity is only given RMSD/TM-score for the author's method itself in the appendix, and there is no direct side-by-side comparison with the main baselines. Therefore, this core conclusion is not closed-loop enough at present.

ATL is slightly overstated. In terms of method definition, ATL actually learns the layer-wise subgraph on the input candidate graph, rather than learning the new topology without constraints. If the key edge is not in the initial candidate graph, ATL cannot be restored. A more accurate expression should be adaptive subtitle selection, not topology learning in a stronger sense.

The gain source of FGD is not strictly isolated. The current ablation shows that FGD is better than vanilla KD, but it can't completely prove that the benefits come from pretrained foundation model prior, not from additional label-related supervision.

In this paper, baselines is re-implemented and iterative generation is removed from R3Design. This may help standardize the protocol, but it may also change the competitiveness of baseline. In terms of efficiency, only the parameters are reported, and the wall-clock/memory index is lacking.

---

> ### Author Rebuttal · Authors · 2026-03-31
>
> We thank the reviewer for the careful reading and constructive comments.
>
> ### 1. On structural fidelity comparison
>
> To strengthen this point, we additionally folded sequences designed by the main baselines under the same folding and evaluation protocol. The resulting RMSD is:
>
> |  Method |   Short  |   Medium  |    Long   |
> | :--: | :--: | :--: | :--: |
> | ATL-FGD | **5.04** | **15.10** | **26.01** |
> | RDesign | 5.22 | 18.54 | 26.94 |
> |  gRNAde | 5.20 | 17.41 | 27.03 |
>
> Due to the rebuttal time limit, we have currently completed folding for these two baselines so far, but this already provides a direct side-by-side comparison under a unified protocol. We will include a more complete structural-fidelity table in the appendix revision where possible.
>
> ### 2. On the scope of ATL and sensitivity to the candidate graph
>
> We appreciate this suggestion. Technically, ATL learns a layer-wise adaptive subgraph over the candidate graph, rather than generating an unconstrained topology from scratch. We will revise the wording accordingly in the paper.
>
> In the additional rebuttal analyses, we examined more about our ATL (please refer to Sec. 3&4 of the X5NH rebuttal and Sec. 1 of the Srni rebuttal for details due to limited rebuttal space).
> Enlarging the candidate graph remains beneficial, which suggests that broader long-range contacts are indeed important.
> However, naive regularization only partially helps and remains clearly weaker than ATL.
>
> We also agree with the reviewer’s point on limitation: if an important contact is absent from the initial candidate graph, ATL cannot recover it.
> Our current results with full-connected candidate graphs suggest that, in practice, ATL is already sufficient to exploit a highly redundant candidate set and learn effective subgraphs.
> That said, learning topology beyond the initial candidate graph, i.e., growing from scratch, is indeed an interesting and valuable future direction.
>
> ### 3. On the source of FGD gains
>
> To address this point, we additionally evaluated a student-FiLM-only variant that keeps the student FiLM head while removing distillation.
> This variant achieves 42.07$\pm$0.59. For more detailed KD analysis, please refer to Sec. 3 of the rebuttal for Reviewer yqSg.
>
> ### 4. On efficiency comparison and R3Design with iterative generation
>
> Our reason for removing iterative generation from R3Design in the current comparison was to standardize all methods under a unified one-pass inference protocol. To address the reviewer’s fairness concern, we additionally evaluated R3Design with iterative generation enabled, which improves recovery to 44.14$\pm$0.27, but with a substantial computational overhead.
>
> | Method | Train Mem (GB) | Train s/epoch | Eval s/epoch |
> |:-:|:-:|:-:|:-:|
> | MPNN baseline | 3.87 | 4.00 | 4.10 |
> | gRNAde | 6.48 | 4.60 | 5.14 |
> | ATL-FGD (k=30) | 4.81 | 4.76 | 4.15 |
> | ATL-FGD (k=64) | 8.93 | 5.68 | 4.22 |
> | ATL-FGD (k=96) | 11.97 | 6.46 | 4.32 |
> | R3Design w/o iter | 14.57 | 5.22 | 5.29 |
> | R3Design w/ iter | 42.13 | 8.20 | 8.80 |
>
> To reduce noise from varying graph sizes across batches, timing results are averaged over 8 epochs, while memory is reported using the peak value over these runs.
> As shown above, our method does introduce additional training memory/time cost as the candidate graph becomes larger, but this overhead is largely linear with k, and the inference overhead remains relatively small.
> In contrast, iterative generation in R3Design yields a modest improvement, but with a large increase in both memory and latency.
> We will update the efficiency discussion beyond parameter count in the revision.

---

> > ### Author Rebuttal · Reviewer_Ndxf · 2026-04-02
> >
> > Thank you for the detailed rebuttal. I appreciate the additional experiments and clarifications, especially on structural fidelity, the scope of ATL, and the efficiency comparison. These responses address several of my concerns and improve my confidence in the work.

---

### Official Review · Reviewer_Srni · 2026-03-17

**Soundness:** 3
**Presentation:** 3
**Significance:** 3
**Originality:** 3
**Overall Recommendation:** 5
**Confidence:** 4

**Summary:**

This paper proposes ATL-FGD, a unified framework for RNA tertiary structure–based sequence design that learns adaptive graph topology and transfers knowledge from RNA foundation models via FiLM-guided distillation.

**Compliance With Llm Reviewing Policy:**

Affirmed.

**Final Justification:**

See rebuttal acknowledgement.

**Key Questions For Authors:**

See above.

**Limitations:**

Yes.

**Strengths And Weaknesses:**

The paper studies an important problem in RNA structure-based design and identifies reasonable limitations in existing approaches such as fixed graph topology and data scarcity. The proposed method is intuitive and shows consistent empirical improvements.

The novelty of this work is somewhat limited. The design of ATL is similar in spirit to existing techniques such as DropEdge/DropMessage, which also manipulate graph connectivity during message passing. It remains unclear whether the observed performance gains come from the adaptive topology learning itself, or simply from alleviating over-smoothing effects. The authors should provide more direct evidence beyond empirical improvements, for example by applying the learned topology from ATL to other baseline models and examining whether it leads to consistent performance gains.

In addition, some important baselines are missing, such as EquiRNA [A] and SE(3)-Hyena [B]. The authors should include comparisons with these methods and discuss them in the related work section. Furthermore, evaluation on larger-scale benchmarks, such as [C], would help better demonstrate the robustness and generalization ability of the proposed method.

- [A] Size-Generalizable RNA Structure Evaluation by Exploring Hierarchical Geometries
- [B] SE(3)-Hyena Operator for Scalable Equivariant Learning
- [C] Beyond Sequence: Impact of Geometric Context for RNA Property Prediction

---

> ### Author Rebuttal · Authors · 2026-03-30
>
> We thank the reviewer for highlighting the importance of the problem and for recognizing the intuitive design and consistent empirical gains of our method.
>
> ### 1. On the relation between ATL and DropEdge / over-smoothing
>
> To directly examine whether the gain mainly comes from generic edge sparsification or alleviating over-smoothing, we conducted an additional control experiment using ATL only (without distillation) and compared it against DropEdge under the same backbone:
>
> | Method | k=64 | k=96 |
> |:-:|:-:|:-:|
> | baseline | 41.76$\pm$0.92 | 41.70$\pm$0.70 |
> | DropEdge | 42.77$\pm$0.33 | 43.44$\pm$1.06 |
> | ATL | **44.21$\pm$0.83** | **45.72$\pm$0.86** |
>
> This comparison suggests two points.
> 1. Generic sparsification does help, indicating that over-smoothing / noisy message passing is indeed a relevant factor.
> 2. ATL remains substantially stronger than DropEdge, indicating that the gain cannot be explained solely by random edge dropping or generic anti-over-smoothing regularization.
>
> We also observe that DropEdge performs particularly poorly on the medium and long subsets (41.65% and 42.31% under k=64), whereas ATL achieves 45.42% and 44.20%, respectively. This suggests that ATL is not merely reducing connectivity, but is learning a more effective layer-wise subgraph than random sparsification.
>
> Overall, over-smoothing is relevant, but the degradation of dense fixed kNN graphs cannot be fully explained by over-smoothing alone. In our setting, irrelevant edges and noisy message passing also matter, and ATL provides a data-dependent way to refine the working graph beyond random edge regularization.
>
> ### 2. On the suggested baselines
>
> We will discuss these works in the related work section. However, they are not directly task-matched baselines for the problem studied here.
>
> - [A] EquiRNA is designed for RNA structure evaluation/scoring, rather than tertiary structure-based sequence design. Its goal is to assess RNA 3D structures, rather than to generate sequences from a target structure.
> - Dataset [C] is designed for RNA property prediction and benchmarks the impact of geometric context on predictive tasks, so it is not directly compatible with our current task setting.
>
> For this reason, we believe these works are valuable for discussion in the broader RNA geometric learning landscape, but they are not directly comparable baselines for the target task of this paper.
>
> Regarding [B] SE(3)-Hyena, we located the Geometric Hyena codebase and started adapting it to the inverse-folding task. However, since the original paper does not directly study inverse folding, additional task-specific adaptation is required for our experimental setting. Due to the limited first-round rebuttal window, we do not yet have reliable results to report, but we will continue exploring this direction.

---

> > ### Author Rebuttal · Reviewer_Srni · 2026-04-03
> >
> > Thank you to the authors for their replies. I believe that with the experimental additions from the rebuttal, the article has achieved logical consistency, therefore I recommend clear acceptance.

---

### Official Review · Reviewer_sDMg · 2026-03-19

**Soundness:** 3
**Presentation:** 3
**Significance:** 3
**Originality:** 3
**Overall Recommendation:** 4
**Confidence:** 3

**Summary:**

This paper introduces a new unified framework named ATL-FGD, for tertiary structure-based RNA design. ATL-FGD has two novel components. In order to capture the flexible and heterogeneous nature of RNA, Adaptive Topology Learning (ATL) to dynamically adjust graph structure at each GNN layer. FiLM-Guided Distillation is introduced to transfer the semantic priors from RNA Foundation Model to the proposed model. Experimental study shows that ATL-FGD achieves consistent improvements on benchmarks

**Compliance With Llm Reviewing Policy:**

Affirmed.

**Key Questions For Authors:**

How sensitive is ATL to the threshold \tau at inference? Is 0.5 fixed for all datasets?

**Limitations:**

Yes

**Strengths And Weaknesses:**

Strengths
1. ATL-FGD targets a meaningful scientific problem. Tertiary structure-based RNA design is a very important problem but currently less mature than protein inverse folding. ATL-FGD is proved to have significant improvements over existing methods on tertiary structure-based RNA design benchmarks.
2. In related protein inverse folding work, pre-trained foundation models are sometimes used directly at inference time. Since such models may have been exposed to benchmark sequences during large-scale pre-training, this can raise concerns about comparison fairness. In contrast, ATL-FGD uses the RNA foundation model only as a teacher during training and transfers its knowledge through distillation. The final model performs inference independently, without relying on the pre-trained model at test time. This design makes the comparison setting cleaner and the reported gains more convincing.
3. The ATL module gives a layer-dependent subgraph without changing the rest of the architecture too much. I like that the mechanism is lightweight and can in principle be dropped into existing GNN backbones.

**Weaknesses**
1. Although the method shows clear improvements over prior baselines on benchmark metrics, the absolute performance is still far from what would likely be required in practical RNA design settings.
2. Tertiary structure-based design ultimately demands sequences that reliably fold into the target structure and satisfy downstream functional constraints, whereas the current results mainly show relative gains on benchmark recovery-style metrics. It remains unclear whether higher sequence recovery leads to meaningfully better structural outcomes. Because multiple sequences can fold to similar RNA structures, better agreement with the reference sequence does not necessarily imply better inverse design quality. The paper would be stronger if it showed that the observed gains translate consistently into improved structural reconstruction compared to other baselines.
3. The paper does not evaluate whether the designed sequences satisfy downstream functional or biophysical constraints. For real RNA design applications, structural plausibility alone is often insufficient. The current study does not assess whether the generated sequences preserve functionality, stability, or experimentally relevant properties, which limits the practical significance of the results.

---

> ### Author Rebuttal · Authors · 2026-03-30
>
> We sincerely thank the reviewer for the careful reading and positive assessment of our work.
>
> ### 1. Regarding gains beyond recovery-style metrics
>
> To examine whether the improvement extends beyond sequence recovery, we additionally evaluated the thermodynamic quality of the designed sequences using RNAfold. ATL-FGD consistently produces more favorable per-nucleotide free energies than RDesign across all length groups:
>
> | Method | Category | MFE/nt | Ensemble FE/nt |
> |:-:|:-:|:-:|:-:|
> | RDesign | Short | -0.09 | -0.10 |
> |  | Medium | -0.27 | -0.29 |
> |  | Long | -0.31 | -0.33 |
> | ATL-FGD | Short | **-0.12** | **-0.13** |
> |  | Medium | **-0.34** | **-0.36** |
> |  | Long | **-0.42** | **-0.44** |
>
> These results suggest that the improvement is not limited to higher recovery, but is also accompanied by improved thermodynamic plausibility of the designed sequences. We will include a more detailed analysis in the appendix in the revision.
>
> ### 2. On the sensitivity to the threshold $\tau$
>
> In the original submission, $\tau=0.5$ is fixed across all experiments. To address this question, we performed an additional sensitivity study under the $k=96$ setting with $\tau \in \{0.3, 0.4, 0.5, 0.6, 0.7\}$. The recovery remains stable across thresholds:
>
> | $\tau$ | Short | Medium | Long | All |
> |:-:|:-:|:-:|:-:|:-:|
> | 0.3 | 45.52% | 45.36% | 45.13% | 45.44% |
> | 0.4 | **45.61%** | 45.87% | 45.30% | **45.65%** |
> | 0.5 | 45.25% | 45.91% | **45.73%** | 45.48% |
> | 0.6 | 44.87% | **46.05%** | 45.70% | 45.29% |
> | 0.7 | 44.88% | 45.35% | 45.16% | 45.04% |
>
> 1. The overall recovery varies only from 45.04% to 45.65%, indicating that ATL is not sensitive to delicate threshold tuning.
> 2. While the overall differences are small, larger thresholds impose stronger edge pruning, and the performance shows a mild trend of first improving and then slightly decreasing.
>
> Overall, we use $\tau=0.5$ as a unified default, since it provides a stable middle-ground setting without heavy tuning. We will include additional analysis under more settings in the ablation studies in the revision.
>
> ### 3. On practical RNA design and downstream constraints
>
> We agree with the comment that real RNA design typically requires additional objectives beyond the inverse-folding backbone, such as functionality, thermodynamic stability, interaction affinity, and other task-specific structural constraints.
> In practical use, ATL-FGD can be naturally combined with a downstream screening framework based on functional scores, thermodynamic indicators, or additional structural criteria. It can also be further combined with iterative screening to perform multi-objective selection with respect to diversity, stability, and downstream constraints.

---

> > ### Author Rebuttal · Reviewer_sDMg · 2026-04-07
> >
> > Thank you for the rebuttal. The additional analyses are helpful and strengthen the paper. In particular, the RNAfold-based free-energy results provide useful evidence beyond sequence recovery alone, and the threshold sensitivity study also helps support the robustness of the ATL component. These additions address part of my original concerns.
> >
> > My main remaining reservation is that the new evidence is still indirect with respect to the ultimate tertiary-structure design objective. Improved thermodynamic plausibility is encouraging, but it does not fully establish that the designed sequences achieve better structural reconstruction quality for the target tertiary structure. Similarly, downstream functional or biophysical constraints are still not directly evaluated, so the practical significance should be stated with some care.
> >
> > I would like to remain my initial evaluation as a weak accept as a result.

---

### Decision · Program_Chairs · 2026-04-30

**Decision:**

Accept (regular)

**Comment:**

In this submission, the authors proposed a new method for boosting tertiary structure-based RNA design. In the rebuttal phase, the authors successfully resolved five reviewers' concerns about implementation details and the solidity of experiments. All the reviewers agreed to accept or weakly accept this work. After reading the paper and the comments, AC decided to accept this submission.